# Ultrafast evolution and transient phases of a prototype out-of-equilibrium Mott–Hubbard material

G. Lantz[1,2], B. Mansart[1], D. Grieger[3], D. Boschetto[4], N. Nilforoushan[1], E. Papalazarou[1], N. Moisan[1], L. Perfetti[5], V.L.R. Jacques[1], D. Le Bolloc'h[1], C. Laulhé[6,7], S. Ravy[1,6], J.-P. Rueff[6,8], T.E. Glover[9], M.P. Hertlein[9], Z. Hussain[9], S. Song[10], M. Chollet[10], M. Fabrizio[3] & M. Marsi[1]

The study of photoexcited strongly correlated materials is attracting growing interest since their rich phase diagram often translates into an equally rich out-of-equilibrium behaviour. With femtosecond optical pulses, electronic and lattice degrees of freedom can be transiently decoupled, giving the opportunity of stabilizing new states inaccessible by quasi-adiabatic pathways. Here we show that the prototype Mott–Hubbard material $V_2O_3$ presents a transient non-thermal phase developing immediately after ultrafast photoexcitation and lasting few picoseconds. For both the insulating and the metallic phase, the formation of the transient configuration is triggered by the excitation of electrons into the bonding $a_{1g}$ orbital, and is then stabilized by a lattice distortion characterized by a hardening of the $A_{1g}$ coherent phonon, in stark contrast with the softening observed upon heating. Our results show the importance of selective electron–lattice interplay for the ultrafast control of material parameters, and are relevant for the optical manipulation of strongly correlated systems.

[1] Laboratoire de Physique des Solides, CNRS, University Paris-Sud, Université Paris-Saclay, 91405 Orsay, France. [2] Institute for Quantum Electronics, Physics Department, ETH Zürich, 8093 Zurich, Switzerland. [3] International School for Advanced Studies SISSA, Via Bonomea 265, 34136 Trieste, Italy. [4] LOA, ENSTA, CNRS, Ecole Polytechnique, F-91761 Palaiseau, France. [5] Laboratoire des Solides Irradiés, Ecole Polytechnique-CEA/SSM-CNRS UMR 7642, 91128 Palaiseau, France. [6] Synchrotron SOLEIL, L'Orme des Merisiers, Saint-Aubin, BP 48, 91192 Gif-sur-Yvette, France. [7] University Paris-Sud, Université Paris-Saclay, 91405 Orsay, France. [8] Sorbonne Université, UPMC Univ. Paris 06, CNRS UMR 7614, Laboratoire de Chimie Physique - Matière et Rayonnement, 11 rue Pierre et Marie Curie, 75005 Paris, France. [9] Advanced Light Source, Lawrence Berkeley National Laboratory, Berkeley, California 94720, USA. [10] LCLS, SLAC National Accelerator Laboratory, Menlo Park, California 94025, USA. Correspondence and requests for materials should be addressed to G.L. (email: lantzg@phys.ethz.ch) or to M.F. (email: fabrizio@sissa.it) or to M.M. (email: marino.marsi@u-psud.fr).

The Mott metal-to-insulator transition (MIT)[1] is the perfect example of how thermodynamic parameters can affect the electronic structure of a material and its conducting properties. At equilibrium, temperature, doping and pressure act as driving forces for such transitions[2], that invariably involve also a lattice modification—either with a change of symmetry, like for instance in $VO_2$ (ref. 3) or with a lattice parameter jump like in $V_2O_3$ (ref. 4). It is actually often unclear whether the lattice or the electronic structure is the trigger for the MIT since at equilibrium both change together. This question can be answered by driving one far from equilibrium and observing the reaction of the other. Thus, time-resolved pump–probe techniques[5–11] can provide this answer, as long as the response of the electrons and of the lattice can be separately analysed.

We adopted a combined experimental and theoretical approach to study the ultrafast evolution of the Mott–Hubbard prototype $(V_{1-x}Cr_x)_2O_3$ (ref. 12). The phase diagram of $V_2O_3$ contains three phases: a paramagnetic metallic (PM) phase, a paramagnetic insulating (PI) phase and an antiferromagnetic insulator phase. The isostructural Mott transition is between the PI and the PM phases[13]. This archetypal material gives the opportunity of comparatively observing the ultrafast evolution of a Mott system starting both from the insulating and the metallic phase, whereas previous studies have generally focused only on the Mott insulator as ground state[5,6,8–10] or, for other systems, on the interplay between spin density wave states[14] and coherent optical lattice oscillations[15]. In all our experiments, the energy of the pump pulses (1.55 eV) corresponds to the transition from $e_g^\pi$ to $a_{1g}$ orbitals. Thus, optical pumping directly increases the $a_{1g}$ population while decreasing the $e_g^\pi$ one. Using time-resolved photoelectron spectroscopy (trPES) we directly probe the electronic structure, while time-resolved X-ray diffraction (trXRD) and time-resolved reflectivity (TRR) give access to the lattice evolution[16–19].

Here we show that with this multitechnique approach one can unambiguously disentangle the contribution of electrons and lattice to the non-equilibrium dynamics of the system, and we find that in the PI phase the gap is instantaneously filled and a non-thermal transient state that lasts 2 ps is created. In the PM phase, the quasiparticle (QP) signal shows an immediate appreciable spectral redistribution across $E_F$, which also lasts 2 ps, once again not compatible with thermal heating. In both phases we find that the lattice conspires to stabilize the non-thermal transient electronic state. *Ab initio* density functional theory with generalized gradient approximations (DFT-GGA) results supplemented by simple Hartree–Fock (HF) calculations suggest that the gap filling is driven by the non-equilibrium population imbalance between the $e_g^\pi$ and $a_{1g}$ orbitals, which effectively weakens the correlation strength.

## Results

**Time-resolved photoelectron spectroscopy**. In vanadium sesquioxide the octahedral crystal field leads to the *d*-orbital splitting into a lower $t_{2g}$ and an upper $e_g^\sigma$. Since the octahedron has a trigonal distortion, the $t_{2g}$ are split into a lower twofold degenerate $e_g^\pi$ orbital and an upper non-degenerate $a_{1g}$ (Fig. 1). The hybridization between the two nearest vanadium atoms, which are lined up along the *c* axis, causes a large splitting between bonding $a_{1g}(\sigma)$ and antibonding $a_{1g}(\sigma^\star)$ states. In spite of that, the $a_{1g}$ orbital remains mostly unoccupied in the PI phase, whereas the $e_g^\pi$ orbitals are occupied by almost one electron each[20,21]. $V_2O_3$ PI can thus be viewed as a half-filled two-band Mott insulator stabilized by the correlation-enhanced trigonal field that pushes above the Fermi energy ($E_F$) the $a_{1g}$ orbitals[21,22], whose occupancy indeed jumps across the doping- or temperature-driven Mott transition[23] causing the opening of a gap[24,25], while

is smoother across the pressure-driven one[13,26]. The nature and indicative energy position of the relevant orbitals for each phase can be found in Fig. 2c,f, where we show the calculated density of states (DOS) from ref. 22. This inequivalent behaviour in temperature versus pressure of the MIT, and the related deep intertwining between strong correlations and lattice structure suggest that a major issue in time-resolved experiments is to distinguish a temperature increase from a transient non-thermal phase, such as hidden phases[27,28].

Before exploring the behaviour of the system after photo-excitation, we present in Fig. 1 the photoemission responses of the PI and PM phases at different temperatures, which give us reference energy distribution curves (EDCs) for the system at equilibrium. In the PM phase the weight near $E_F$ increases with decreasing temperatures, which is consistent with the expected behaviour of the QP[25,29]. In the PI phase, the temperature increase fills the gap, which is consistent with the results from Mo *et al.*[30]. The temperature difference between 200 and 220 K, $\Delta T = 20$ K, matches the estimated temperature rise brought by the pump laser pulse for the fluence used in our pump–probe photoemission experiments (see Supplementary Notes 1 and 2 and Supplementary Figs 1–3). Therefore, the difference curves between high and low-temperature spectra at fixed doping may serve to compare the non-equilibrium spectra with reference to the thermal ones.

The non-equilibrium electron dynamics has been studied with pump–probe photoemission. The differences between positive and negative time delays are shown in Fig. 2a–c for the PI phase. As representative of the time evolution, we consider the timescan at $-0.1$ eV below $E_F$ (Fig. 2a), whose decay can be fitted with two exponentials. The details on the fitting procedure can be found in the Supplementary Note 3 and Supplementary Fig. 4. The first exponential with a $76 \pm 6$ fs decay time is limited by our time resolution, corresponds to the hot electron relaxation after photoexcitation and clearly indicates a strong electron-phonon coupling. We associate the second longer timescale of $1.7 \pm 0.3$ ps with the lifetime of a transient state, as suggested by comparing the EDCs at 50 fs, 400 fs and 2 ps with the thermal differences at equilibrium (black). At 50 fs delay (red curve) an increase in spectral weight is clearly visible both below and above $E_F$, an evidence of creation of in-gap states. The EDC cannot be fitted with a Fermi-Dirac distribution, since the system is still strongly out-of-equilibrium. The 400 fs delay spectrum has instead no weight above $E_F$: the excess electrons have cooled down. Nevertheless, the spectrum still deviates from the equilibrium one, in particular at $-0.1$ eV binding energy, indicating that, even though the electrons have relaxed, the state is different from the thermal configuration. A spectral difference equivalent to the thermal state at equilibrium can instead be found after 2 ps, when the transient state has fully relaxed.

Figure 2d–f reports the photoexcited behaviour of pure $V_2O_3$ (PM) at the same fluence of 1.8 mJ cm$^{-2}$. The timescan at 0.1 eV above $E_F$ (Fig. 2d) shows a fast decay with a characteristic time of $70 \pm 6$ fs (limited by the time resolution) and a slower one of $1.8 \pm 0.4$ ps, similar to the time constants found in the PI phase. Indeed, the EDC differences at 50 and 400 fs delays are compatible with the hot electrons not being thermalized at 50 fs and almost thermalized at 400 fs.

The observed spectral changes obtained around $E_F$ by keeping the sample at $T$ and photoexciting with a pump pulse cannot be ascribed to heating, but rather to a genuine non-thermal transient state[31,32]. In particular, both spectra at 50 and 400 fs (Fig. 2e) suggest that there is more weight both below and above $E_F$ in the photoexcited state at temperature $T$ than in the equilibrium state at $T + \Delta T$. Therefore, the reduction of density of states around $E_F$

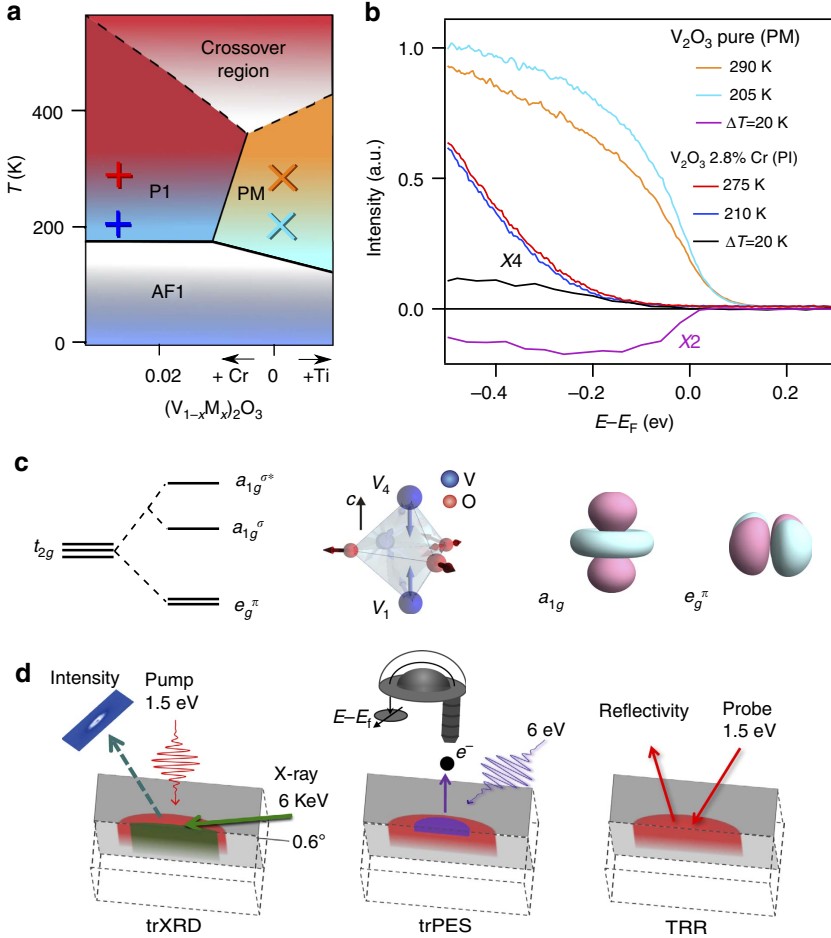

**Figure 1 | Equilibrium properties of $(V_{1-x}M_x)_2O_3$ and experimental techniques used.** (**a**) $(V_{1-x}M_x)_2O_3$ phase diagram; the cross and the plus symbols indicate the experimental data points. (**b**) Temperature dependence of the equilibrium photoemission spectra for $(V_{1-x}Cr_x)_2O_3$ ($x = 0.028$ PI phase, $x = 0$ PM phase). Temperature differences are shown for each doping, which are used as a thermal equilibrium reference in comparison with the photoexcited spectra. Upon increasing the temperature, the spectral weight is transferred into the Mott gap in the PI phase, whereas the QP peak weakens in the PM phase. (**c**) Representation of the orbital splitting and their geometry. (**d**) Schematic of the experiments using an optical pump and different probes: trXRD, trPES and TRR.

is not compatible with a thermally excited configuration. This non-thermal state relaxes in 2 ps, similarly to the PI phase.

**Time-resolved reflectivity.** Further evidence in support of a transient non-thermal phase comes from the lattice. In Fig. 3 we present TRR measurements that provide information on the transient response of the fully symmetric $A_{1g}$ optical phonon, which corresponds to the breathing of one entity of $V_2O_3$ as shown in Fig. 1. Consistently with previous studies[33,34], we observe an electronic excitation peak lasting about 200 fs, similar to the trPES response observed in Fig. 2. The succeeding coherent oscillations can be analysed by Fourier transform, which is compared in Fig. 3b,c with the $A_{1g}$ mode measured with Raman spectroscopy at equilibrium. Surprisingly, the mode displays a blue-shift of up to 14% compared with the equilibrium frequency for both PI and PM phases. Such a blue-shift, that is, a phonon hardening, is certainly non-thermal in nature. Indeed a temperature increase causes instead softening and consequently a red-shift[35]. Hardening of the $A_{1g}$ phonon actually corresponds to a decrease of the average distance between the two closest vanadium atoms, $d(V_1 - V_4)$[20]. It should be underlined that this coherent phonon hardening is present for both the PM and PI phases, and that its decoherence time is about 2 ps: these features

are in full agreement with the behaviour observed for the electronic degrees of freedom with trPES (Fig. 2). There is consequently a strong evidence of a transient phase that does not correspond to any equilibrium phase of the system, involving both the electronic and lattice structure and present in both PM and PI phases.

**Time-resolved X-ray diffraction.** In order to verify our interpretation on the nature of this transient phonon blue-shift, we performed a trXRD study on the same crystals used for the trPES and TRR measurements. The effective fluence was only slightly higher than in trPES because the probing depth of XRD is much higher than PES: since the behaviour of the trPES signal is linear versus fluence (see Supplementary Fig. 5) the results among the different experimental methods can be safely compared because we are in the same excitation regime.

In Fig. 3d,e we present the time-dependent intensity of the Bragg reflections (116) and (204) for the PI phase. The peak positions do not change until 4 ps, when the lattice parameters start being modified by the onset of the acoustic wave (as discussed in Supplementary Note 4 and Supplementary Fig. 6). Here we focus on the behaviour during the first few

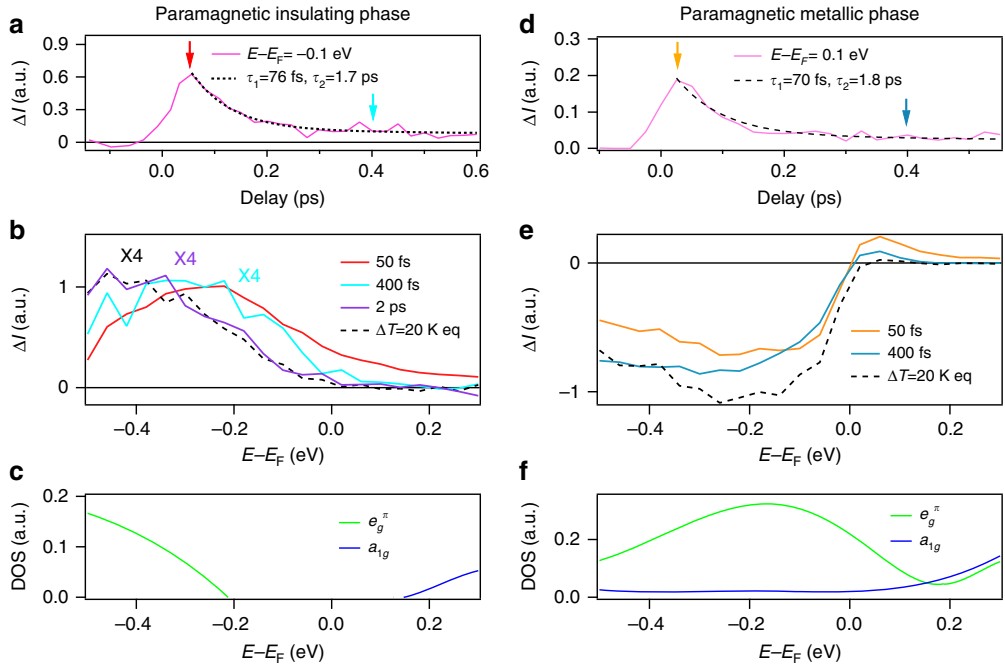

**Figure 2 | trPES for $(V_{1-x}Cr_x)_2O_3$ for the PI and PM phases at a fluence of 1.8 mJ cm$^{-2}$.** (a) Time evolution of the intensity difference at $-0.1$ eV, the curve is fitted with a double exponential. (b) PES intensity difference for $\Delta t = 50$ fs, 400 fs and 2 ps are shown for the PI phase as well as the equilibrium temperature difference from Fig. 1. The 50 and 400 fs differences show that the spectral weight is transferred inside the Mott gap, differently from a purely thermal effect. This non-thermal distribution relaxes within 2 ps. (c) Orbital character of the DOS near $E_F$ extracted from ref. 22. (d–f) Same as (a–c) but for the PM phase. The time evolution is fitted with a double exponential for the energy above $E_F$.

picoseconds, when the intensities of both Bragg reflections vary, while the lattice parameters are constant. Supposing that the symmetry of the crystal stays the same, the diffracted intensity can be simulated by a change of the vanadium Wyckoff position, $Z_V$, and a Debye–Waller factor[18], while keeping the $V_1$—$O_1$ distance constant. The change of the oxygen Wyckoff position affect the peak intensity of less than 0.02% for the (116) and about 1% for the (024) peak. The distance of the nearest vanadium atoms is given by the relation $d(V_1 - V_4) = (2Z_V - 0.5)c$, where $c$ is the lattice constant. The (116) and (024) structure factors vary in opposite directions with $Z_V$. We find that, $d(V_1 - V_4)$ goes from 2.744 Å to a minimum value of 2.71 Å before 1 ps ($d(V_1 - V_4)_{PM} = 2.69$ Å). The Debye–Waller factor is responsible for only 0.1% of the intensity change before 4 ps. After 4 ps, due to the lattice expansion, the changes in structure factor are no longer sufficient to explain the experimental curves because the peak position also starts changing (see Supplementary Fig. 6). The trXRD response was not able to resolve the coherent lattice oscillations, due to limits in the signal-to-noise levels attainable during the measurements, but it does confirm that the blue-shift in the coherent phonon frequency is related to a transient reduction of the average distance $d(V_1 - V_4)$. By comparing the temporal evolution of the different experimental results, the TRPES measurements show that the electronic structure is modified faster, and that the lattice deformation follows—which is expected for a prototype Mott system. The resulting non-thermal state is visibly more metallic in the PI phase, and seems most likely more delocalized in the PM one as well. In both cases, this state is stabilized by a transient lattice deformation that shortens the distance between the two nearest vanadium atoms and consequently increases the covalent bonding between the $a_{1g}$ orbitals. The fact that trXRD gives a slightly longer relaxation time with respect to trPES can be explained by the different probing depths of the two techniques[6,36].

## Discussion

In $V_2O_3$ the $e^\sigma$ orbital lie around 3 eV above $E_F$ (ref 20,22; see Supplementary Fig. 7). Therefore, the most favourable transition with a 1.5 eV optical excitation is the transition from $e_g^\pi$ to $a_{1g}$, which is dipole-active. Figure 2c,f shows the orbital nature of the bands near $E_F$, which are affected by the pump pulse. We considered a three-band Hubbard model at one-third filling for the $t_{2g}$ orbitals with the tight-binding hopping parameters of ref. 20, and analysed this model by means of the HF approximation[22] using as control parameter, after a Legendre transform, the occupancy difference between $e_g^\pi$ and $a_{1g}$ orbitals. In order to describe an insulator within an independent particle scheme as HF we had to allow for magnetism; our insulator is thus closer to the antiferromagnetic insulator phase low-temperature phase rather than to the high-temperature PI[22]. Within HF, the total energy, shown in Fig. 4a, has two minima, a stable one at $n_{a_{1g}} \simeq 0.5$ describes the insulator and a metastable minimum at $n_{a_{1g}} \simeq 0.625$ that instead represents a metal. In Fig. 4b we plot the density of states for three different values of $n$, two in the insulating phase and one in the metal. We modelled the experiment in the PI phase starting from a Slater determinant that describes the HF insulator with a number of electrons transferred from the valence band of mostly $e_g^\pi$ character to the conduction one, with $a_{1g}$ character, and studied its time evolution within time-dependent HF.

We find it is enough to transfer $\sim 0.13$ electrons to the conduction band to drive the system into the metastable metallic phase, as pictorially drawn in Fig. 4a, which is consistent with the experimental excitation that are 8% for a fluence of 8 mJ cm$^{-2}$ in the trXRD and TRR experiments and 3.1% for the trPES. In other words, the non-thermal phase appears in this theoretical scenario as a metastable state that pre-exists in equilibrium and can be nucleated within the stable insulator through the photoexcitation. Since time-dependent HF does not account for dissipation, we

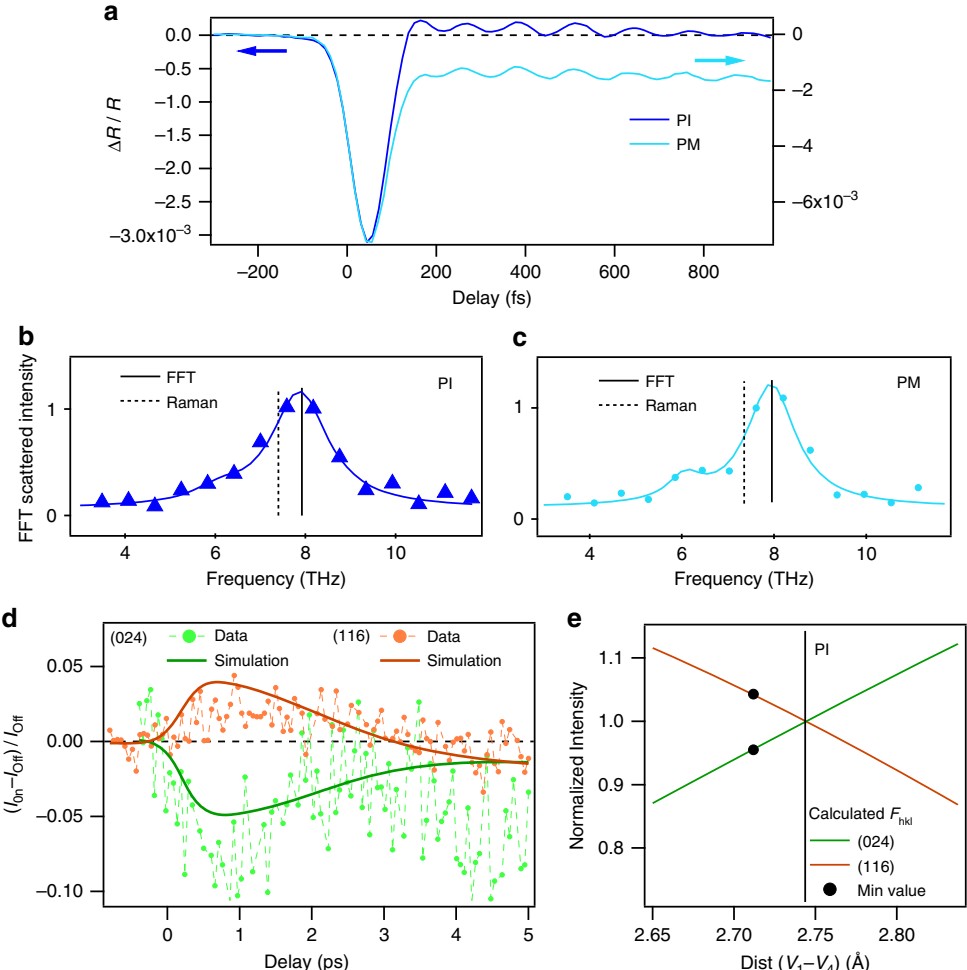

**Figure 3 | Experimental results from TRR and trXRD.** (**a**) TRR traces for $(V_{1-x}Cr_x)_2O_3$ ($x = 0.028$ PI and $x = 0$ PM) for a fluence of 8 mJ cm$^{-2}$: the $A_{1g}$ coherent phonon is clearly visible. (**b,c**) Fast Fourier transform of TRR traces compared with equilibrium Raman spectroscopy for the PM phase and PI phase, respectively. The $A_{1g}$ pump–probe frequencies (full) present a clear blue-shift compared with the equilibrium frequency (dashed) in both phases. (**d**) trXRD measurements in the PI phase for a fluence of 8 mJ cm$^{-2}$, showing the pump–probe diffraction peak intensities for the Bragg reflections (116) and (024). The solid lines are the simulation as explained in text. (**e**) The calculated structure factor versus the shortest vanadium distance ($V_1 - V_4$). The black dots represent the minimum distance observed extracted from **d**.

cannot describe the subsequent break-up of the metastable metal nuclei back into the stable insulator, which experimentally occurs after few ps. The $a_{1g}$ orbitals being bonding, an overpopulation would bring the nearest vanadium atoms together. A LDA + U calculation with such an overpopulation of the $a_{1g}$ orbitals is able to capture the observed phonon hardening (see Supplementary Note 5 and Supplementary Figs 8–10).

With a combined experimental and theoretical approach, we show that the ultrafast response of the prototype Mott–Hubbard compound $(V_{1-x}Cr_x)_2O_3$ is characterized by a non-thermal transient phase in which the system remains trapped before relaxing to the final thermal state. The formation of this non-thermal phase is very fast for both PM and PI—faster than our experimental time resolution—and it is eminently electronic in nature, being driven by a transient overpopulation of a bonding $a_{1g}$ orbital. A selective lattice deformation, strikingly highlighted by the $A_{1g}$ phonon hardening, further stabilizes this non-thermal transient phase, whose lifetime grows up to few ps: this direct comparative analysis of the evolution of the metallic and insulating phases is relevant for all the efforts aiming at photoinducing phase transitions in correlated materials, including possible technological applications like ultrafast switches. Our

results thus show that a selective electron–lattice coupling can play an important role in out-of-equilibrium Mott systems, even though the main actor remains the strong correlation, and appear to be of very general validity, suggesting that similar non adiabatic pathways can be found in other multiband Mott compounds following excitation with ultrafast light pulses.

## Methods

**Samples.** All the $(V_{1-x}Cr_x)_2O_3$ samples used in our experiments are high-quality single crystals from Purdue University. They were oriented using Laue and X-ray diffraction, and cut along the (001) plane. For both the X-ray diffraction and the TRR measurements the samples were mechanically polished in order to have a flat surface. For all this specimens we could consistently observe nice coherent phonon oscillations, in agreement with previous studies[33], which indicates a good crystal quality and rules out spurious effects in comparison with photoemission results. For the photoemission experiments the samples were cleaved along the (001) plane, where the QP photoelectron yield is most pronounced for the metallic phase[25].

All time-resolved measurements were performed at 200 K.

**Time-resolved photoelectron spectroscopy.** trPES measurements were performed on the FemtoARPES set-up[37]. A Ti:sapphire laser delivers 35 fs, 1.58 eV pulses that are split in two: one part is used to generate the fourth harmonic for the ultraviolet photoemission probe the rest serves for the pump pulses to excite the material. The original repetition rate is 250 kHz but it was reduced by a factor four

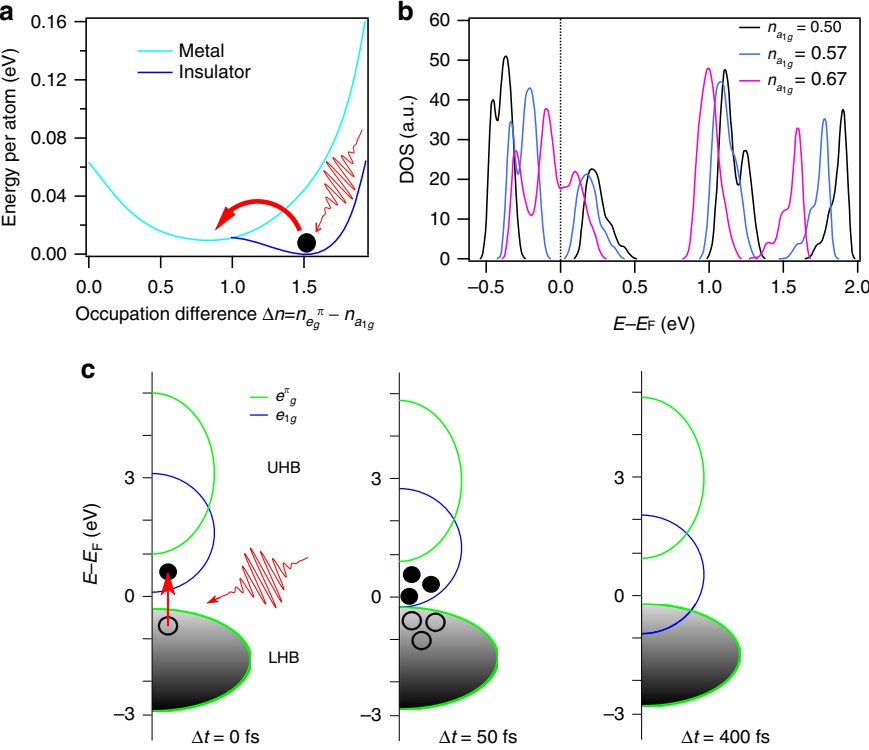

**Figure 4 | Theoretical calculations.** (**a**) Hatree–Fock total energy as function of the occupancy difference between $e_g^\pi$ and $a_{1g}$ orbitals (the total occupancy is 2). (**b**) DOS for different occupancies of the $a_{1g}$. (**c**) Schematic view of the proposed mechanism involved in the photoexcitation of a Mott material, where the $a_{1g}$ states lower in energy both of the PM and PI phases.

by using a chopper in order to avoid residual heat. The energy resolution is better than 70 meV and the time resolution is better than 80 fs. The photoemission spectra were taken around $\Gamma Z^{25}$. The $(V_{1-x}Cr_x)_2O_3$ samples were oriented to have the [001] direction in the hexagonal notation perpendicular to the surface, and along the analyser axis. The samples were cleaved *in situ* in order to obtain a clean surface.

**Time-resolved reflectivity.** The TRR experiments were performed with a 1 kHz Ti:sapphire laser, which delivers 45 fs, 1.55 eV pulses. A near-normal incidence geometry was used and the pump and probe beams were cross-polarized. The background was subtracted before performing the Fast Fourier transform (FFT). The experiments were performed at 200 K.

**Time-resolved X-ray diffraction.** trXRD measurements were performed with sub-ps time resolution at the x-ray pump probe (XPP) end-station of the Linac coherent light source[38]. The incidence angle for the 8 keV X-ray beam was 0.6°, while for the optical laser beam it was 12°: this geometry allowed us to match the penetration depths and retain a temporal resolution of the order of 200 fs. The estimated penetration depth for the X-rays is 120 nm, whereas it is 88 nm for the optical laser. The sample was cooled down to 200 K with a cryo-jet. The different Bragg reflections were observed using a two-dimensional detector.

The Bragg peak intensity was measured by integrating over a $20 \times 20$ pixels wide region centred around the peak. Due to monochromatization of the X-ray beam, any energy jitter from the self amplified spontaneous emission (SASE) process results in an X-ray intensity fluctuation on the sample. The flux and the position of the incident photons were measured by intensity-position monitors. A key point in analysing the data was choosing an intensity and position range, which optimizes the signal-to-noise ratio: this was done by analysing the Bragg reflection behaviour for negative time delays. The best results were obtained by cutting off the 20% lowest shots and the 5% highest, as well as filtering on the positions that deviate more than one s.d. in $x$ and $y$. The data were then corrected for the delay-time jitter using the LCLS timing-tool. We chose a time bin of 50 fs, which gives an average of $\sim 10^6$ photons per delay.

**Data availability.** The data that support the findings of this study are available from the corresponding authors upon reasonable request

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

## Acknowledgements

This work has been partly supported by the EU under the contract Go Fast (Grant No. 280555) and under the ERC project No. 692670. G.L., N.M., L.P., E.P. and M.M. acknowledge financial support by Investissement d'Avenir Labex PALM (ANR-10-LABX-0039-PALM), by the Equipex ATTOLAB (ANR11-EQPX0005-ATTO-LAB) and by the Région Ile-de-France through the programme DIM OxyMORE. D.B. acknowledges the financial support of the French Procurement Agency (DGA) of the French Ministry of Defense. The Advanced Light Source is supported by the Director, Office of Science, Office of Basic Energy Sciences, of the U.S. Department of Energy under Contract No. DE-AC02-05CH11231. Use of the Linac Coherent Light Source (LCLS), SLAC National Accelerator Laboratory is supported by the U.S. Department of Energy, Office of Science, Office of Basic Energy Sciences under Contract No. DE-AC02-76SF00515.

## Author contributions

G.L., E.P. and L.P. performed the trPES experiments. B.M., E.P. and D.B. performed the TRR experiments. G.L., D.B., E.P., N.M., V.L.R.J., D.B., C.L., S.R., J.-P.R. and M.M. carried out the trXRD experiments, with the contribution of T.E.G., M.P.H., Z.H., S.S. and M.C. N.N., G.L., N.M. and V.L.R.J. performed the trXRD analysis. D.G. and M.F. carried out the theoretical calculations and interpretation. G.L., M.F. and M.M. wrote the article, with inputs from all the authors. M.M. conceived and coordinated the project.

## Additional information

**Competing financial interests:** The authors declare no competing financial interests.

**Publisher's note**: 

