## [Peer Review File · Nature Communications]

Reviewers' comments:

Reviewer #1 (Remarks to the Author):

This is a thorough presentation combining three time-resolved experimental techniques as well as theoretical approaches to study the Mott-Hubbard material V₂O₃ in its pure and Cr-doped form, i.e., the paramagnetic insulating and metallic phases, respectively. The conclusions of this work are well-grounded in the comprehensive sets of data provided and would be very appropriate for Nature Communications, as this is an important model system that has somehow had little attention from ultrafast techniques so far. Multiple lines of evidence point to a non-thermal excitation lasting a few picoseconds, driven first by electronic excitation. The quality of the series of experiments is high and they are generally well-explained. However, each of the experiments has issues, of varying degrees, that must be fixed before publication can be recommended.

Time-resolved Photo Electron Spectroscopy:

The thermal data used to compare with the photoexcited data, i.e., "Delta T = 20 K," is an important point but is described very tersely in the main text. It took some searching thru the Supplemental Materials to realize it was not an interpolation between measurements at ~200 and 290 K (the symbols in Fig. 1a, which are said to represent the experimental data points). Thus, I would suggest that the sentence, in the middle of the second column of page 2, that introduces Delta T be changed to something like this: "The temperature difference between 200 and 220 K, Delta T = 20 K, matches the estimated temperature rise..."

In the fits to the trPES data, the fitted values of the tau₁ time constant are not really significant since (as we find out from the Supplemental Materials) they basically match the time resolution. I don't disagree that this feature of the curve corresponds to the hot electron relaxation, but it is misleading to quote a time of 76 fs without explaining that is simply the resolution of the measurement.

Figures 2c and 2f appear without any mention in the text. (I am also not sure what it means that they are "extracted from" reference 16.) If these plots are worth including in the figure, then they deserve at least a brief discussion.

Coherent Phonon Spectroscopy:

This is just time-resolved reflectivity (TRR) but to call it CPS assumes that the signal from a coherent phonon will be measured. I do not think that the name of an experimental technique should change depending on the outcome of the measurement. (That is, the same measurement would certainly not have been called CPS if oscillations were not observed.) Therefore, this series of measurements should simply be referred to as TRR, as the authors did in previous publications (e.g., ref. 28).

Time-resolved X-Ray Diffraction:

Regarding Fig. 3d, the fit is fairly poor after ~3.5 ps. Presumably this may be related to the lattice expansion beginning at this time, but the authors should discuss this or seek to improve the fits in this time frame.

In the simulations of the diffraction intensities, the authors should not ignore motion of the oxygen atoms. First of all, this is because V is a fairly light element, and there are two and a half times more O than V; the total numbers of electrons of the V and O atoms are actually pretty close. But more specifically, allowing the O to move will improve the fit in Fig. 3d. If the Wyckoff position x for the O

atoms increases by $\sim 1.5\%$, this will decrease the intensity of the 024 reflection by a few percent, without affecting the 116 intensity. This will likely improve the fit in the 0 to 2 ps range. It will also serve to maintain the length of the V(1)-O(1) bond, which in the authors' current model (no O motion) would shrink by over one percent.

I also find Fig. 3 of the Supplemental Materials difficult to interpret. What do the X and Y axes refer to? Do they correspond to particular directions in reciprocal space, or just the axes of the area detector? And what is the inset supposed to show? Why does the change in intensity appear so asymmetric in the areas where the intensity increases vs. where it decreases? There could be some interesting science here, but without any discussion it's impossible to tell.

There are a few other issues which need attention. There are a few misspellings, such as the last word of the abstract. A few references are incorrect ([1] should be volume 49, page 72) or incomplete ([17]). And in the Supplemental Materials, in the last sentence of the first paragraph, it seems that it should say that all time-resolved measurements were performed at 200 K rather than just "all measurements."

Reviewer #2 (Remarks to the Author):

The manuscript by Lantz et al., reports the new framework to control the non-equilibrium states behind the Mott metal-insulator transition on the prototypical compound V₂O₃ by the selective lattice deformation due to the optical pump pulse. The concept and methodology are pretty interesting with the combined time-resolved experiments probes both the electronic structures and the lattice deformation. Therefore, I am positive to recommend for the publication in the nature communications. However, at least in the present form, there are several issues to be clarified to convince the author's statement before the publication as listed below.

(1)

To fit the time evolution of the intensity difference of the photoemission spectral weight at -0.1 eV, the author introduce the double exponential function having two time constants. However, the experimental time-evolution curves seem to show the simple decay and may be possible to be fitted by the simple exponential function. The author should carefully discuss to exclude the possibility of the artificial fitting at least in the supplementary materials. Moreover, it could be helpful to show the two components of exponential functions together with the fitted double exponential function.

(2)

The author stress that it is possible to select the excitation path from the eg_{π} to a_{1g} excitation in the pump pulse. However, it is not clearly explained why the only a_{1g} states are activated instead of the eg_{σ} states. Since the tr-XRD and CPS experiments support the excitation to the a_{1g} states, the conclusion may not be changed, but to improve the readability it is needed to discuss the reasoning of the "selective" a_{1g} excitation explicitly. For example, if one use the energy level diagram based on the Rodolakis et al. PRL(2010) [20] gives the 10 Dq of 2.0 eV with trigonal field of 0.3 eV, supporting the a_{1g} excitation. However, if one refers the 10 Dq value of 1.0 eV by J. H. Park et al. PRB 61, 11506 (2000) [21] the situation may change that it is also possible to excite to the eg_{σ} states. For this, It may be helpful to refer the paper by Saha-Dasgupta et al. arXiv:0907.2841v1, which could be reference [17] in the manuscript.

(3)

The DOS plotted in Fig.2c from reference [16] shows the gap opening between the eg_{pi} and $a1g$ state. However, the spectral function for the PI phase in ref. [16] show that the gap is almost closed, which is consistent with hard-X-ray photoemission spectrum obtained in the PI phase by Fujiwara et al., PRB 84 075117 (2011) though the photoemission does not probe the gap but only probes the occupied side of the one-electron removal spectral function. Anyway, one cannot find the same DOS plot in Fig.2c in the ref.[16], so that the author should note proper reference.

(4)

The surface treatment is different depending upon the experiments, i.e., cleaving for the tr-PES, and polishing for the tr-XRD. Even if the photon-in and photon-out measurements, the distortion induced by the polishing may give the artifact due to the stress to the crystals. For this point, it should be noted in the technical part in the supplementary material whether it is no problem or not. At least XRD measurements could be possible to measure for both polished and cleaved samples to mention the above issue.

(5)

The laser power of the tr-PES measurements (1.8 mJ/cm^2) is much smaller than that for the tr-XRD and CPS measurements (8 mJ/cm^2). The difference of the fluence is factor 4, which is still twice as large for the difference of the equivalent effective fluence between the tr-PES and tr-XRD experiments as discussed in the supplementary material, and may induce the different mode between the tr-PES and others. The author should carefully discuss this issue.

Other minor issues are listed below,

(6)

In Fig.1 (a), the author put the cross marks indicating the experimental temperature, but are they placed on the correct temperature ??

(7)

In the supplementary materials, the author discuss the quasi particle (QP) "peak" for tr-PES spectra especially in the section of the "temperature dependence" and the figure caption of Fig. 1(b). However, there is no "peak" structures beside the Fermi level as discussed in the high energy photoemission as S. K. Mo et al. PRL 90, 186403 (2003), Panacione et al. PRL 97, 116401 (2006), and H. Fujiwara et al.. PRB 84, 075117 (2011). In such a sense, it could be better to use the quasi particle "weight" or "component".

(8)

The author mentioned the tr-PES covers the electronic structure around the Gamma point, but how does the author select the k-points especially for the three-dimensional materials as V2O3 without assuming the inner potential ?

(9)

There is a typo in the sentence at the line 13 from the bottom in page 3. "of states around EF is "Inot"...": Inot -> not .

Reviewer #3 (Remarks to the Author):

The study by Lantz et al. comprises a set of measurements on the prototypical Mott insulator V2O3

concerning the ultrafast dynamics of electrons and phonons after interband photoexcitation. The central finding of the paper is that the frequency of an A_{1g} coherent phonon mode increases for a few ps immediately upon photodoping while it decreases when the material is thermally heated. This observation is interpreted as a new transient phase of the material which is transferred into a local minimum of the hyperpotential surface due to excitation of electrons into a band formed by bonding orbitals. The claim is supported by additional experiments based on transient photoemission and X-ray scattering. This manuscript is written in a clear form and contains original results on the very active field of nonequilibrium dynamical properties of strongly correlated electron systems. The content is clearly of interest for the readership of Nature Communications. This referee recommends publication after the following minor points have been considered in a revised version:

1) Data in Fig. 2a and d: Here, two important timescales are deduced which are of central importance for the physical discussion in the paper. Therefore, it would be more efficient to plot the data on a log scale in order to clearly identify the exponential decay components. Also, it appears that there is some negative offset of the signal at negative delay times at least in Fig. 2a. How did the authors treat this fact, especially when deducing the 1.7 ps time constant where the signal-to-noise ratio is relatively small?

2) In the motivational part of their paper, the authors cite mostly work that has been performed with the same methods as the ones applied in their study. Especially, "coherent phonon spectroscopy" (= impulsive stimulated Raman scattering) is credited to give access to the lattice evolution. The authors should bear in mind that direct measurements of the complex dielectric function are now also possible in the mid to far infrared and with sub-cycle resolution of the electric field, giving complementary information on low-energy degrees of freedom as compared to Raman techniques and tendentially more details due to the inherently two-time character. For example, the authors state that "time-resolved pump-probe techniques can provide this answer, as long as the response of the electrons and the lattice can be separately analyzed." But there is no reference to Phys. Rev. Lett. 99, 116401 (2007) and/or Phys. Rev. B 83, 195120 (2011) where simultaneous probing of the ultrafast dynamics of electronic and lattice degrees of freedom in a strongly correlated system has been achieved for the first time and in the mixed Mott-Peierls system VO₂ which can be regarded as a benchmark for the case of V₂O₃. Also, it is stated that previous studies on photoexcitation of materials with energy gaps due to strong electronic correlations "have generally focused only on the insulator as ground state". In fact, the authors should acknowledge that a periodic re-opening of the spin-density-wave gap with the A_{1g} phonon frequency after excitation of the metallic phase of BaFe₂As₂ has been reported in Nature Materials 11, 497 (2012).

REVIEWERS' COMMENTS:

Reviewer #1 (Remarks to the Author):

The authors have carefully revised the manuscript, and I feel it is an excellent paper ready for publication in Nature Communications.

HOWEVER, I would ask that the authors take one more pass thru the Supplementary Materials to improve the organization. Specifically, the text does not refer to the figures in order and does not refer to Figure 2 at all.

Also, please be consistent in capitalization of trXRD vs. trXrd.

Reviewer #2 (Remarks to the Author):

The comments raised from my side is appropriately revised, so that I would like to recommend to Nature communications.

Reviewer #3 (Remarks to the Author):

In their revised manuscript, the authors have corresponded adequately to the criticism by the reviewers. This referee now suggests rapid publication in Nature Communications.

Dear Editors,

enclosed please find a revised version of our manuscript "Ultrafast evolution and transient phases of a prototype out-of-equilibrium Mott-Hubbard material". We thank all the referees for their competent reports and for their constructive criticism. We have carefully taken into account all their remarks, as detailed here below. We believe this has improved the quality of our paper, and we do hope it will meet your satisfaction.

Kind regards,

The Authors

Reviewer #1 (Remarks to the Author):

This is a thorough presentation combining three time-resolved experimental techniques as well as theoretical approaches to study the Mott-Hubbard material V₂O₃ in its pure and Cr-doped form, i.e., the paramagnetic insulating and metallic phases, respectively. The conclusions of this work are well-grounded in the comprehensive sets of data provided and would be very appropriate for Nature Communications, as this is an important model system that has somehow had little attention from ultrafast techniques so far. Multiple lines of evidence point to a non-thermal excitation lasting a few picoseconds, driven first by electronic excitation. The quality of the series of experiments is high and they are generally well-explained. However, each of the experiments has issues, of varying degrees, that must be fixed before publication can be recommended.

We thank the referee for his/her careful and critical review of our manuscript, for its overall positive evaluation, and for the very constructive and competent remarks, which really improve our paper. We provide here below our replies to the points raised:

R1-A) Time-resolved Photo Electron Spectroscopy:

R1_1) The thermal data used to compare with the photoexcited data, i.e., “Delta T = 20 K,” is an important point but is described very tersely in the main text. It took some searching thru the Supplemental Materials to realize it was not an interpolation between measurements at ~200 and 290 K (the symbols in Fig. 1a, which are said to represent the experimental data points). Thus, I would suggest that the sentence, in the middle of the second column of page 2, that introduces Delta T be changed to something like this: “The temperature difference between 200 and 220 K, Delta T = 20 K, matches the estimated temperature rise...”

The curve shown for delta T=20 K is indeed a real measurement and not an interpolation. As suggested, we have changed the text to make it more clear.

R1_2) In the fits to the trPES data, the fitted values of the tau1 time constant are not really significant since (as we find out from the Supplemental Materials) they basically match the time resolution. I don't disagree that this feature of the curve corresponds to the hot electron relaxation, but it is misleading to quote a time of 76 fs without explaining that is simply the resolution of the measurement.

This time constant is indeed limited by our time resolution. We followed the referee's suggestion and we now say that it is limited by the resolution in the text.

R1_3) Figures 2c and 2f appear without any mention in the text. (I am also not sure what it means that they are “extracted from” reference 16.) If these plots are worth including in the figure, then they deserve at least a brief discussion.

The purpose of these figures is to help the reader follow the nature of the orbitals involved in our pump-probe experiment, and the fact that electrons are mainly excited into the a_{1g} levels. We have replaced the curves from ref. 16 (now ref 21) with curves published by some of us in Phys. Rev. B 92, 075121 (2015), and as suggested we make reference to them in the text.

R1_B) Coherent Phonon Spectroscopy:

R1_4) This is just time-resolved reflectivity (TRR) but to call it CPS assumes that the signal from a coherent phonon will be measured. I do not think that the name of an experimental technique should change depending on the outcome of the measurement. (That is, the same measurement would certainly not have been called CPS if oscillations were not observed.)

Therefore, this series of measurements should simply be referred to as TRR, as the authors did in previous publications (e.g., ref. 28).

We agree with this remark, we have changed the name to TRR in the text.

R1_C)Time-resolved X-Ray Diffraction:

R1_5) Regarding Fig. 3d, the fit is fairly poor after ~ 3.5 ps. Presumably this may be related to the lattice expansion beginning at this time, but the authors should discuss this or seek to improve the fits in this time frame.

We thank the referee for this remark, we made this point clear in the text. Indeed, since the peak is moving starting around 4 ps, we are not anymore on the maximum of the rocking curve with the detector after 4 ps. The loss of intensity is also due to the fact that the peak is moving. We only look at one cut in the reciprocal space and the integrated intensity is proportional to the structure factor only if the peak does not move. The acoustic effects are also interesting and will be the subject of another publication.

R1_6) In the simulations of the diffraction intensities, the authors should not ignore motion of the oxygen atoms. First of all, this is because V is a fairly light element, and there are two and a half times more O than V; the total numbers of electrons of the V and O atoms are actually pretty close. But more specifically, allowing the O to move will improve the fit in Fig. 3d. If the Wyckoff position x for the O atoms increases by $\sim 1.5\%$, this will decrease the intensity of the 024 reflection by a few percent, without affecting the 116 intensity. This will likely improve the fit in the 0 to 2 ps range. It will also serve to maintain the length of the V(1)-O(1) bond, which in the authors' current model (no O motion) would shrink by over one percent.

We thank the referee for this remark. Indeed, the (116) is not much affected by the oxygen but the (024) is. We have redone the simulations keeping the V-O bond constant and replaced them in the paper. It does give a better fit of the experimental data, and confirms the physical implications of our observations.

R1_7) I also find Fig. 3 of the Supplemental Materials difficult to interpret. What do the X and Y axes refer to? Do they correspond to particular directions in reciprocal space, or just the axes of the area detector? And what is the inset supposed to show? Why does the change in intensity appear so asymmetric in the areas where the intensity increases vs. where it decreases? There could be some interesting science here, but without any discussion it's impossible to tell.

The x and y axis are the axes of the detector. The inset is the detector image difference between laser on and off. The difference is asymmetric because the peak is shifting at the same time as decreasing in intensity. After 4 ps, the image is also not at the maximum of the rocking curve therefore its interpretation is not straightforward. The acoustic part of the dynamics will be subject of another publication. We have changed the legend of the figure for the axis and removed the inset because it was misleading and unnecessary.

R1_8) There are a few other issues which need attention. There are a few misspellings, such as the last word of the abstract. A few references are incorrect ([1] should be volume 49, page 72) or incomplete ([17]). And in the Supplemental Materials, in the last sentence of the first paragraph, it seems that it should say that all time-resolved measurements were performed at 200 K rather than just "all measurements."

We have corrected the mistakes, we thank the referee for pointing them out.

Reviewer #2 (Remarks to the Author):

The manuscript by Lantz et al., reports the new framework to control the non-equilibrium states behind the Mott metal-insulator transition on the prototypical compound V₂O₃ by the selective lattice deformation due to the optical pump pulse. The concept and methodology are pretty interesting with the combined time-resolved experiments probes both the electronic structures and the lattice deformation. Therefore, I am positive to recommend for the publication in the nature communications. However, at least in the present form, there are several issues to be clarified to convince the author's statement before the publication as listed below.

We thank the referee for his/her detailed and competent analysis of our paper, and for the appreciation expressed for our efforts and results. We hereby provide our replies to the various, very pertinent remarks:

Reviewer #2 (Remarks to the Author):

R2_1) To fit the time evolution of the intensity difference of the photoemission spectral weight at -0.1 eV, the author introduce the double

exponential function having two time constants. However, the experimental time-evolution curves seem to show the simple decay and may be possible to be fitted by the simple exponential function. The author should carefully discuss to exclude the possibility of the artificial fitting at least in the supplementary materials. Moreover, it could be helpful to show the two components of exponential functions together with the fitted double exponential function.

The double exponential is needed as we can see that the spectra from the 400 fs and 2 ps are clearly different. We have added a figure to explain in detail the fitting procedure in the supplementary.

R2_2) The author stress that it is possible to select the excitation path from the eg_{π} to $a1g$ excitation in the pump pulse. However, it is not clearly explained why the only $a1g$ states are activated instead of the eg_{σ} states. Since the tr-XRD and CPS experiments support the excitation to the $a1g$ states, the conclusion may not be changed, but to improve the readability it is needed to discuss the reasoning of the “selective” $a1g$ excitation explicitly. For example, if one use the energy level diagram based on the Rodolakis et al. PRL(2010) [20] gives the $10 Dq$ of 2.0 eV with trigonal field of 0.3 eV, supporting the $a1g$ excitation. However, if one refers the $10 Dq$ value of 1.0 eV by J. H. Park et al. PRB 61, 11506 (2000) [21] the situation may change that it is also possible to excite to the eg_{σ} states. For this, It may be helpful to refer the paper by Saha-Dasgupta et al. arXiv:0907.2841v1, which could be reference [17] in the manuscript.

There are indeed some discrepancies in the literature on the value of the octahedral field $10Dq$. We followed the referee’s suggestion to quote the paper by Saha-Dasgupta et al, which support our own calculations that place the eg_{σ} states 3 eV above Fermi. We have added a paragraph in the discussion about the selective excitation.

R2_3) The DOS plotted in Fig.2c from reference [16] shows the gap opening between the eg_{π} and $a1g$ state. However, the spectral function for the PI phase in ref. [16] show that the gap is almost closed, which is consistent with hard-X-ray photoemission spectrum obtained in the PI phase by Fujiwara et al., PRB 84 075117 (2011) though the photoemission does not probe the gap but only probes the occupied side of the one-electron removal spectral function. Anyway, one cannot to find the same DOS plot in Fig.2c in the ref.[16], so that the author should note proper reference.

In our experiment the gap is 0.2 eV same as in *Phys. Rev. B* **74**, 165101 (2006) where they use hard X-ray photoemission (same group as in PRB 84 075117 (2011)). It should be emphasized that in a previous low energy

ARPES study of ours, Rodolakis et al. PRL 102, 066805 (2009), Ref. 26 - hence in experimental conditions much more similar to this work - the opening of a gap of about 0.2 eV is also clearly visible at the PI-PM transition (Fig. 1 in ref. 25).

The plot in Fig 2 f was in ref[16] (now ref[21]) however Fig. 2c was extracted from Fig 12 in *Phys. Rev. B* **92**, 1–14 (2015). We now extracted both from *Phys. Rev. B* **92**, 1–14 (2015).

R2_4) The surface treatment is different depending upon the experiments, i.e., cleaving for the tr-PES, and polishing for the tr-XRD. Even if the photon-in and photon-out measurements, the distortion induced by the polishing may give the artifact due to the stress to the crystals. For this point, it should be noted in the technical part in the supplementary material whether it is no problem or not. At least XRD measurements could be possible to measure for both polished and cleaved samples to mention the above issue.

The fact that we consistently see coherent phonon oscillations in TRR means that the crystal quality of the polished surfaces was good and rules out spurious effects. In the PES experiment, the quasiparticle temperature behavior is also a sign of a good surface. When the surface is bad no temperature behavior can be seen. We discuss this point in the supplementary material.

R2_5) The laser power of the tr-PES measurements (1.8 mJ/cm^2) is much smaller than that for the tr-XRD and CPS measurements (8 mJ/cm^2). The difference of the fluence is factor 4, which is still twice as large for the difference of the equivalent effective fluence between the tr-PES and tr-XRD experiments as discussed in the supplementary material, and may induce the different mode between the tr-PES and others. The author should carefully discuss this issue.

As discussed in the supplementary, there is a factor 2 between the effective fluences. We have added a figure in the Supplementary with the fluence dependence of the trPES experiment. The behavior is linear for these fluences, therefore this suggest that we are in the same excitation regime for all the experiments. We now discuss this point in the main paper as well.

----- Other minor issues are listed below,

R2_6) In Fig.1 (a), the author put the cross marks indicating the experimental temperature, but are they placed on the correct temperature ??

The cross are the experimental data for Fig 1- b). The positions were a bit too high, we modified them.

R2_7) In the supplementary materials, the author discuss the quasi particle (QP) "peak" for tr-PES spectra especially in the section of the "temperature dependence" and the figure caption of Fig. 1(b). However, there is no "peak" structures beside the Fermi level as discussed in the high energy photoemission as S. K. Mo et al. PRL 90, 186403 (2003), Panacione et al. PRL 97, 116401 (2006), and H. Fujiwara et al.. PRB 84, 075117 (2011). In such a sense, it could be better to use the quasi particle "weight" or "component".

We replaced the expression "quasiparticle peak" by "quasiparticle weight".

R2_8) The author mentioned the tr-PES covers the electronic structure around the Gamma point, but how does the author select the k-points especially for the three-dimensional materials as V2O3 without assuming the inner potential ?

We thank the referee for pointing this out. indeed it is not the Gamma point but a point along GammaZ which yields a strong quasiparticle signal, as discussed in detail in Rodolakis, F. *et al. Phys. Rev. Lett.* **102**, 66805 (2009).

R2_9) There is a typo in the sentence at the line 13 from the bottom in page 3. "of states around EF is "lnot"...": lnot -> not .

Corrected

Reviewer #3 (Remarks to the Author):

The study by Lantz et al. comprises a set of measurements on the prototypical Mott insulator V2O3 concerning the ultrafast dynamics of electrons and phonons after interband photoexcitation. The central finding of the paper is that the frequency of an A1g coherent phonon

mode increases for a few ps immediately upon photodoping while it decreases when the material is thermally heated. This observation is interpreted as a new transient phase of the material which is transferred into a local minimum of the hyperpotential surface due to excitation of electrons into a band formed by bonding orbitals. The claim is supported by additional experiments based on transient photoemission and X-ray scattering. This manuscript is written in a clear form and contains original results on the very active field of nonequilibrium dynamical properties of strongly correlated electron systems. The content is clearly of interest for the readership of Nature Communications. This referee recommends publication after the following minor points have been considered in a revised version:

We thank the referee for his/her review and for the positive evaluation of our manuscript. We revised the manuscript following his/her indications as detailed here below:

R3_1) Data in Fig. 2a and d: Here, two important timescales are deduced which are of central importance for the physical discussion in the paper. Therefore, it would be more efficient to plot the data on a log scale in order to clearly identify the exponential decay components. Also, it appears that there is some negative offset of the signal at negative delay times at least in Fig. 2a. How did the authors treat this fact, especially when deducing the 1.7 ps time constant where the signal-to-noise ratio is relatively small?

We added a figure in the supplementary with the two components and the residual. The delays 400 fs and 2 ps show clearly a difference around -0.1 eV which highlights the need for 2 timescales. The negative offset was just noise, we added more range before time zero.

R3_2) In the motivational part of their paper, the authors cite mostly work that has been performed with the same methods as the ones applied in their study. Especially, "coherent phonon spectroscopy" (= impulsive stimulated Raman scattering) is credited to give access to the lattice evolution. The authors should bear in mind that direct measurements of the complex dielectric function are now also possible in the mid to far infrared and with sub-cycle resolution of the electric field, giving complementary information on low-energy degrees of freedom as compared to Raman techniques and tendentially more details due to the inherently two-time character. For example, the authors state that "time-resolved pump-probe techniques can provide this answer, as long as the response of the electrons and the lattice

can be separately analyzed." But there is no reference to Phys. Rev. Lett. 99, 116401 (2007) and/or Phys. Rev. B 83, 195120 (2011) where simultaneous probing of the ultrafast dynamics of electronic and lattice degrees of freedom in a strongly correlated system has been achieved for the first time and in the mixed Mott-Peierls system VO₂ which can be regarded as a benchmark for the case of V₂O₃. Also, it is stated that previous studies on photoexcitation of materials with energy gaps due to strong electronic correlations "have generally focused only on the insulator as ground state". In fact, the authors should acknowledge that a periodic re-opening of the spin-density-wave gap with the A_{1g} phonon frequency after excitation of the metallic phase of BaFe₂As₂ has been reported in Nature Materials 11, 497 (2012).

We have added the reference Phys. Rev. Lett. 99, 116401 (2007) since it is very similar to V₂O₃. We stated that previous experiments on Mott materials have been carried out mostly on insulator. We also added the reference Nature Materials 11, 497 (2012) for the interplay between spin density wave and coherent phonons.